# Use of Thidiazuron for High-Frequency Callus Induction and Organogenesis of Wild Strawberry (*Fragaria vesca*)

**DOI:** 10.3390/plants10010067

**Published:** 2020-12-30

**Authors:** Hsiao-Hang Chung, Hui-Yao Ouyang

**Affiliations:** Department of Horticulture, National Ilan University, Yilan 26047, Taiwan; oyalex29@yahoo.com.tw

**Keywords:** wild strawberry, *Fragaria vesca*, plant regeneration, plant growth regulator

## Abstract

Strawberry, belonging to the *Fragaria* genus, is an important crop that produces popular fruits globally. *F. vesca*, known as wild strawberry, has great characteristics, such as a robust and powerful aroma, making it an important germplasm resource. The present study aims to establish an efficient regeneration method for the in vitro propagation of *F. vesca*. Firstly, leaf explants were used to induce callus formation on a Murashige and Skoog medium with combinations of cytokinins (thidiazuron (TDZ) and 6-benzylaminopurine (BA)) and auxin (2,4-dichlorophenoxyacetic acid (2,4-D)). Among them, 0.45–4.54 µM TDZ combined with 0.45–4.53 µM 2.4-D exhibited a high induction rate after 4 weeks of culturing. Different explants were examined for their ability to form a callus, and whole leaves on the medium containing 2.27 µM TDZ and 2.27 µM 2,4-D showed the highest callus induction rate at 100% after 2 weeks of culturing in darkness. The highest shoot regeneration ability through organogenesis from the callus was obtained at 0.44 µM BA. After 2 weeks of culturing, the shoot regeneration rate and shoot number per explant were 96% and 19.4 shoots on an average, respectively. Rooting of shoots was achieved using indole-3-butyric acid (IBA) or an α-naphthaleneacetic acid (NAA)-containing medium, and the resulting plantlets were acclimatized successfully and formed flowers eventually. In this report, we demonstrated that shoot organogenesis was derived from the meristematic cells of calli and by transferring the induced calli to a 0.44 µM BA medium, the regeneration period can be shortened greatly. A protocol for the efficient regeneration of wild strawberry was established, which might be useful for their large-scale propagation or obtaining transgenic plants in the future.

## 1. Introduction

Strawberry, classified in the genus *Fragaria* (Rosaceae family), is one of the most economically important and popular crops worldwide for the fresh fruit market, as well as in the fruit processing industry, such as jams, juices, and jellies. Due to their delicate flavor, aroma and nutritional value, the demand for strawberries has dramatically increased in the past decades [1]. According to the latest data from the Food and Agriculture Organization of United Nation, the annual world production of strawberries in 2018 had reached 8.3 million tons, which is almost double amount compared to 2000 (4.5 million tons) [2].

The nutritional composition and relevant bioactive compounds of strawberries have been well characterized [3]. Their high vitamin C content has been established, where a serving of 10 strawberries can provide 95% of the recommended dietary requirement [1]. The anthocyanin contents of various strawberries range from 150 to 800 mg/kg of fresh weight [4,5]. Other flavonoids, such as catechin, and derivatives of quercetin and kaempferol were also found in various strawberry species [6,7]. Regarding the health-promoting properties of strawberries, the antioxidant abilities from different cultivars have been reported [7]. Many studies also demonstrated the potential health benefits of anthocyanins, including being antioxidative, anticancer, antidiabetic and anti-obesity, as well as preventing cardiovascular disease and improved visual health, etc. [8,9,10,11,12,13]. *Fragaria × ananassa* (octoploid) and its derived varieties are the most important commercial cultivated strawberry, which is derived from the hybrid of *F. chiloensis* (octoploid) and *F. virginiana* (octoploid) [1]. However, there are still many wild and ancient species with distinct characteristics. The wild species *F. vesca* (diploid), which was used as the plant material in this study, has a much stronger aroma than the current cultivated types [14]. Olefinic monoterpenes and myrtenyl acetate are the major volatile compounds from *F. vesca* fruits, but these could not be found in the cultivated species [15]. Therefore, the wild *F. vesca* species is an important germplasm resource for strawberry breeding in the future.

Conventionally, the propagation of strawberry is achieved by a vegetative approach using runners, which is labor intensive, timing consuming and can cause the transmission of viral diseases [16]. Therefore, the healthy plantlets form micropropagation technology has become one of the best substitutions. Several studies revealed that micropropagated strawberry plants have better characteristics, including crown size, runner number, flowering time and yield of berries [17,18,19]. Many studies on the micropropagation of strawberries have been reported, which demonstrated various morphological development, such as callus induction, shoot regeneration, rooting, protoplast culture and somatic embryogenesis [20,21,22,23]. Adventitious shoot organogenesis is the most common regeneration pathway in strawberries. Different explants and plant growth regulators (PGRs) have been applied on various genotypes. Previous studies demonstrated that different cultivars have a distinct response to PGRs, explants and environmental factors [16,20,24]. Thus, each cultivar might need to be characterized under different factors and conditions for the most efficient regeneration rate. Thidiazuron (TDZ) was first reported by Arndt et al. [25] as a cotton (*Gossypium hirsutum*) defoliant. Because of its efficient role in plant tissue culture, TDZ has been widely use during the past decades and shown both auxin- and cytokinin-like effects [26]. Several reports on strawberry regeneration also indicated the high efficiency of TDZ [20,24,27,28], which was considered as a great PGR for in vitro morphogenesis studies.

In vitro culturing is one of the most important techniques for the production of pathogen-free and true-to-type certified plantlets, and also for the genetic improvement of berries by advanced biotechnologies. The aim of this study was to establish an efficient and reliable system using TDZ for inducing calli by the dedifferentiation of explants and shoot organogenesis. Different auxins were used for rooting, eventually obtaining viable plantlets of *F. vesca*. The high frequency of callus induction was not only good for mass propagation, but also provided great materials for the genetic transformation. The established regeneration system in this study can be utilized for genetic studies in the future.

## 2. Results

### 2.1. Callus Induction from Whole Leaf Explants under Combination Treatments of Cytokinins and 2,4-Dichlorophenoxyacetic Acid (2,4-D)

Whole strawberry leaves were used as the explants, and different combinations of growth regulators, i.e., thidiazuron (TDZ) or 6-benzylaminopurine (BA) plus 2,4-D, were used to induce the callus. The results of the 4-week culture are shown in Table 1. Whether under light or darkness, the explants on hormone-free or 2,4-D-only media did not form any callus or other morphogenesis and showed a higher browning rate. The explants on cytokinin-only media had a lower induction rate and down to 0% on the medium with a high TDZ concentration (4.54 µM). The best callus induction rates were obtained in the combinations of TDZ 2.27 µM and 2,4-D of 0.45–4.53 µM, with an average of 97.3% in darkness and even up to 100% callus induction in the light condition (Table 1). Through the microscopic observation, it was observed that the callus tissues appeared compact (Figure 1A), and many calli were induced from the leaf edge (Figure 1B).

However, in the treatments of BA and 2,4-D, only the combinations with a higher 2,4-D concentration showed better callus induction rates, such as BA 2.22 µM + 2,4-D 4.53 µM, BA 4.44 µM + 2,4-D 2.27 µM and BA 8.88 µM + 2,4-D 2.27 µM, which achieved an average induction rate of 88% in darkness; a 92% callus induction was obtained on the media of BA 2.22 µM + 2,4-D 4.53 µM in the light condition. It was also found that the explants had a very high browning rate on BA-only, 2,4-D-only and BA +2,4-D combinations in the light (Table 1). Regarding the regeneration of buds in this study, no adventitious shoots and roots were formed after the 4-week treatment, whether in the darkness or light (data not shown). Above all, the results showed that the combination of TDZ and 2,4-D would induce callus regeneration more efficiently than the combination of BA and 2,4-D. Browning occurred more severely in the light condition, but TDZ could dramatically reduce this phenomenon. Therefore, TDZ is a more suitable cytokinin-like PGR for callus induction of wild strawberry. All combinations of TDZ and 2,4-D showed an extremely high induction rate in both the light and dark conditions, with no significant difference between the combinations. It indicated that we should shorten the culture period to find the optimized PGR concentration.

### 2.2. The Effect of Different Plant Explants under 2,4-D and TDZ Combinations

Due to the previous study, we observed the callus regeneration firstly after 2 weeks of culturing (Figure 2A,B) in this part of the work. A similar result was obtained, in that the combination of cytokinin TDZ and auxin 2,4-D can induce more calli than those without adding any growth regulators and using cytokinin or auxin alone. We observed that the callus induction rate in whole leaf explants reached 100% only after 2 weeks in the medium with TDZ 2.27 µM and 2,4-D 2.27 µM in the dark (Table 2). The higher TDZ (4.54 µM) dose combined with 2,4-D 0.45 µM treatment also exhibited a high induction percentage of 92%. For tip blade explants, the greatest callus formation rate was 96%, found in TDZ 2.27 µM and 2,4-D 0.45 µM in the dark. Regarding the basal blade, the treatments containing both TDZ and 2,4-D showed a high callus induction from 72–88% in the dark. Relatively, the efficiency of the callus induction was lower when the petioles were used as explants and no shoot regeneration was found in all treatments after 2 weeks of culturing. The results showed that leaves are better explants for inducing calli, especially whole leaves and basal blades, and, in general, the dark condition was more suitable than light.

After 4 weeks of culturing on the same medium, calli and a few shoots were found (Figure 2C–F). The 100% callus induction could be also achieved from whole leaves treated with TDZ 4.54 µM + 2,4-D 0.45 µM, the basal blades treated with TDZ 2.27 µM + 2,4-D 0.45 µM and TDZ 4.54 µM + 2,4-D 0.45 µM, and the petioles treated with TDZ 2.27 µM + 2,4-D 2.27 µM in darkness (Table 3). Basically, the high induction rate of calli could be observed by adding TDZ 0.45–4.54 µM and 2,4-D 0.45–4.53 µM after 4 weeks of culturing in the dark, especially the whole leaf and basal blade explants, which showed no significant difference between combinations with both TDZ and 2,4-D PGRs. The trend of this result was similar to that in Section 2.1. In this test, we also found the shoot regeneration occurred in the medium to which TDZ 2.27 µM + 2,4-D 0.45 µM and TDZ 4.54 µM + 2,4-D 0.45 µM were added, though the regeneration rate was only 12–16% in the whole leaf and basal blade explants (Table 4) and the shoot number was only 0.8–1.0 shoot per explant (Table 5). This part of the results demonstrated that the combination mediums with TDZ and 2,4-D were great for callus induction and proliferation, but not suitable for shoot regeneration.

### 2.3. Shoot Regeneration from Calli Using Different Cytokinin Treatments

For the shoot regeneration, the calli were used as explants in this study for a 2-week culture. In general, more shoots could be induced in light than in darkness. The callus treated with BA 0.44 µM under the light condition gave the highest shoot induction rate of 96%, though it was not statistically higher than the treatment of TDZ 0.45 µM and kinetin (KIN) at all concentrations at the second week (Table 6). However, the number of induced buds was 19.4 per explant, which was significantly higher (*p* < 0.05) than all the treatments, except KIN 2.33 µM. This efficient regeneration rate will be an ideal treatment for mass propagation in the future. Although a high induction rate was found for the TDZ 0.45 µM treatment in the dark, the pale green color of the buds was not considered to be as healthy as those in the light condition. By histological analysis, the meristematic cells were observed on the edge of the callus explants (Figure 3A,C). These cells exhibited a smaller size and obviously were more densely stained when compared to the original callus cells. The leaf primordium and shoot apical meristem that developed from the meristematic cells were also observed (Figure 3B), which revealed the adventitious shoot organogenesis from the calli of wild strawberry (Figure 3D). No somatic embryogenesis was found in this study.

### 2.4. Root Induction from Shoot Explants Using Different Auxins

In the study of root induction, we found that IBA showed a better induction rate than NAA. A higher browning rate and lower root induction rate were observed under the light condition, demonstrating that the dark condition is more suitable for root induction. In darkness, the rooting rates of IBA 0.41 µM, IBA 4.14 µM and IBA 8.28 µM were 12%, 8% and 4%, respectively (Table 7). After 8 weeks of culturing, better induction ratios and root numbers were observed in the IBA 0.41 µM treatment under the dark condition, followed by the IBA 1 mg/L treatment (Figure 4).

### 2.5. Plant Acclimatization

The 5 cm in size strawberry plants, with an intact root system, were moved out from the in vitro system (without the remaining medium) and planted in three-inch pots containing a mixed medium of peat soil, pearlite and vermiculite, with the ratio of 1:1:1 in volume. The strawberry plants were then acclimated in a low-light shading house for two weeks. The acclimatized strawberries were transferred into the five-inch pots after one month. The strawberries without the formation of a running stem developed white flowers (Figure 5).

## 3. Discussion

Regarding the in vitro morphogenesis of strawberry, both somatic embryogenesis and organogenesis have been described in different strawberry cultivars [21]. Different factors, including explants (e.g., runner, nodal, stem and leaf), plant growth regulators (e.g., BA, TDZ, NAA, IBA and 2,4-D) and environmental conditions, have been reported. Wild strawberry (*F. vesca*) was the first domesticated strawberry in Europe and acquired popularity in the 1500s and 1600s [1]. Although it is an old species, several great characteristics, such as a robust and powerful aroma, makes it a great source for strawberry breeding in the future. Therefore, we established a serial regeneration method for wild strawberry in this study.

The present research used leaves as the explant, which are the most abundant tissue of this plant and thus less harmful to it when excised. In the first test, the combination of two cytokinins (TDZ and BA) and one auxin (2,4-D) was applied for the regeneration of wild strawberry. We observed that only callus tissues were formed after 4 weeks culturing and the efficiency of TDZ was better than BA (Table 1). The browning rate of the leaf explant was high in the light condition without adding TDZ. Cappelletti et al. [29] showed that the callus induction rate of *Fragaria × ananassa* “Calypso” reached 100% in the combination of TDZ 1.0 mg/L + 2,4-D 0.2 mg/L and TDZ 0.5 mg/L + 2,4-D 0.02 mg/L after 42 days. Our wild strawberry gave a similar result to “Calypso”, and our induction rate reached 100% after 28 days. In general, one of the key factors for regeneration is to utilize a combination of cytokinin and auxin in the culture medium. Cappelletti et al. [29] have successfully induced the shoots at an 80% regeneration rate in the “Sveva” cultivar, by using IBA 3.0 mg/L and BA 0.2 mg/L. In the “Chandler” cultivar, the combination of 2,4-D 1.0 mg/L and BA 0.1 mg/L in the MS medium was able to prompt the callus formation rate up to 90% [30]. Biswas et al. [31] showed that application of NAA 4.0 mg/L and BA 1.5 mg/L in three strawberry (*Fragaria* spp.) clones could induce the callus formation from 72 to 89% using different explants, including leaves, runners and nodal segments. According to the previous studies above, different concentrations of PGRs and different strawberry cultivars showed variability in the regeneration rate, indicating that each genotype has specific requirements for regeneration.

The type of explant was also an important factor for successful plant regeneration. A broad range of explants have been characterized in strawberry, e.g., leaves [32,33,34], petioles [33,35], stems [36], peduncles [37], stolons [37], stipules [35,38], roots [35,38], anthers [39] and sepals [40]. Whole leaf, tip blade, basal blade and petiole tissues were also examined for callus induction. In general, the four types of explants exhibited a high induction percentages with TDZ and 2,4-D in the dark culture, especially after 4 weeks of culturing (Table 3). However, for the tip blade, the induction rate dropped to 48–60% when a higher dose of TDZ (4.54 µM) was applied and the browning phenomenon occurred prevailingly in tip blade explants at all PGR combinations (data not shown). Shoot organogenesis also could be observed in a few combinations. For callus induction from leaves, the best and quicker condition was found in the medium with TDZ 2.27 µM and 2,4-D 2.27 µM, incubated in the dark. Shoots will form if the culture is kept in the same condition. However, a larger number of shoots was obtained in the following treatment.

For shoot regeneration from callus tissue, Rugini and Orlando [38] found that 84–96% of callus explants from *Fragaria × ananassa* Duch. induce 4–8 shoots per explant in the medium containing BA and IBA. In our experiment, three types of cytokinins, i.e., TDZ, BA and KIN, were applied. In contrast with callus induction, higher shoot regeneration rates were observed in the light condition. After a 2 week culture, the regeneration rate of 96% with 19.4 shoots per explant was observed in BA 0.44 µM (Table 6) and the number of shoots increased to 52 after 4 weeks of culturing (data not shown), which is the optimum PGR concentration in this part of the work. Compared with the shoot regeneration rate in the callus-inducing medium after a 4-week culture (Table 4), only 4–12% was observed. Less efficient results were seen in root induction, with a 12% induction rate in the IBA 0. 41 µM medium (Table 7).

In conclusion, we established a simple and efficient regeneration protocol for wild strawberry (*F. vesca*) by using leaf explants. Calli of wild strawberry can be formed in 2 weeks from leaf explants in a medium containing TDZ 2.27 µM and 2,4-D 2.27 µM in the dark. Then, the calli were cultured on a shooting medium with BA 0.44 µM for another 2 weeks in the light condition. A number of shoots can be obtained within a month. Sarker et al. [28] showed that the shoots of *F. vesca* cv. Baron Solemacher were regenerated from leaf explants via the callus stage after 6–9 weeks in the same medium. Therefore, to shorten the regeneration period, application of a shooting medium could be essential. This protocol can be useful for rapid and large-scale shoot propagation and might be applied for the generation of transgenic plants in the future. However, regarding rooting, it still should be improved by applying other types of auxin or environmental conditions.

## 4. Materials and Methods

### 4.1. Plant Material and In Vitro Seed Germination

Wild strawberry (*Fragaria vesca*) seeds were purchased from Known-You Seed Co. Ltd., Kaohsiung, Taiwan. Seeds were sterilized by soaking in a 0.5% NaClO (sodium hypochlorite) solution with 10 mg L^−1^ Tween-20 (Sigma-Aldrich Inc, St. Louis, MO, USA) for 10 min and then rinsed with 75% ethanol for 1 min. Finally, all seeds were washed 3 times with sterile ddH_2_O to remove any traces of ethanol and dried on a sterilized Petri dish. Sterilized seeds were placed on 1/2 MS medium (Duchefa Biochemie B.V., Haarlem, Netherlands) [41] containing 30 g L^−1^ sucrose (Taiwan Sugar Corp., Tainan, Taiwan) and 3 g L^−1^ phytagel (Sigma-Aldrich Inc, St. Louis, MO, USA) at pH 5.7, which was autoclaved at 121 °C at 15 psi pressure for 30 min in 500 mL flasks. All seeds and seedlings were incubated in a growth chamber under a 16/8 h (light/dark) photoperiod at a photon flux density of 28–36 μmol m^−2^s^−1^ (daylight fluorescent tubes FL-30D/29, 40 W, China Electric Co., Taipei, Taiwan) and temperature of 25 ± 2 °C. In vitro seedlings were subcultured in the same medium every 2 months.

### 4.2. Effect of 2,4-D and Cytokinins on Callus Induction Using Whole Leaf Explants

Fully expanded leaves (around 1 cm) were excised from in vitro seedlings (about 3–4 cm in height) as explants. For induction of calli, the leaf explants were cultured under different PGR combinations and light conditions. The combinations of 2,4-D (0, 0.45, 2.27, 4.53 µM) with TDZ (0, 0.45, 2.27, 4.54 µM) or BA (2.22, 4.44, 8.88 µM) were added to the 1/2 MS basal medium mentioned above. The leaf explants were placed on the culture medium in 90 × 15 mm^2^ Petri dishes under darkness or light conditions at the same photoperiod and intensity described above. All explants were cultured at a temperature of 25 ± 2 °C. Five dishes (replicates), each containing 5 explants, were utilized in this experiment. Data were collected after 4 weeks.

### 4.3. Effect of Explants on Callus Induction

Different explants, i.e., whole leaves, leaf tip blades, leaf basal blades and petioles, were excised from in vitro seedlings and placed on the 1/2 MS basal medium supplemented with the PGR combination of 2,4-D (0, 0.45, 2.27 µM) and TDZ (0, 0.45, 2.27, 4.54 µM). The dark and light conditions described above were also examined for different explant types and the temperature was kept at 25 ± 2 °C. Five 90 × 15 mm^2^ Petri dishes (replicates), each containing 5 explants, were used in this experiment. Data were collected after 2 and 4 weeks.

### 4.4. Effect of Different Cytokinins on Shoot Regeneration Using Callus Explants

For preparing callus explants, one of the best PGR combinations (2.27 µM TDZ and 2.27 µM 2,4-D) for callus induction was applied using whole leaf explants. After 4 weeks of culturing, the induced calli were cut into small clumps (around 0.5 cm^3^) and incubated on the 1/2 MS basal mediums containing TDZ (0, 0.45, 2.27, 4.54, 9.08 µM), BA (0.44, 2.22, 4.44, 8.88 µM) or KIN (0, 0.47, 2.33, 4.65, 9.30 µM). The dark and light conditions were also examined. Five 55 × 15 mm^2^ Petri dishes (replicates), each containing 5 explants, were used in this experiment. Data were collected after 2 weeks.

### 4.5. Root Induction

After shoot regeneration, the shoots were examined for the root induction rate using different auxins and concentrations. Auxins, IBA (0, 0.41, 1.03, 2.07, 4.14, 8.28 µM) and NAA (0.48, 1.20, 2.40, 4.80, 9.60 µM), were added to the 1/2 MS medium. Five shoot explants were incubated in a 125 mL flask as 1 replicate and 5 replicates were applied for each treatment. The effects of the dark and light conditions were examined in the study. Data were collected after 8 weeks.

### 4.6. Acclimatization of Regenerated Plants

After 6 months of root induction, plantlets around 5 cm in height with well-developed shoots and roots were transplanted into 3-inch plastic pots with potting mix (peat moss:vermiculite:perlite = 1:1:1). The plantlets were incubated in a shaded greenhouse under a natural photoperiod and temperature for acclimatization.

### 4.7. Histological Analysis

For histological observation, tissue samples were fixed in FPA (formaldehyde:propionic acid:glycerol:95% ethanol:distilled water = 1:1:3:7:8 in volume) solvent for 48 h and dehydrated through a tertiary butyl alcohol series. After removing the water, samples were infiltrated with liquid paraffin at 65 °C in the oven and embedded in paraffin wax. The embedded tissue wax blocks were trimmed and sectioned at 10 μm using a rotary microtome (Leica RM2125 RT, Nussloch, Germany). The tissue sections were serially mounted on glass slides and stained with 1% safranin-O (Sigma-Aldrich Inc., St. Louis, MO, USA) for 30 min followed by 5% fast green (Sigma-Aldrich Inc., USA) for 10 s. After washing out the residual stain, sections were permanently mounted on slides for observation.

### 4.8. Statistical Analysis

All experiments in this study were designed with a complete randomization of the PGR combinations. The morphogenesis of the tissue explants was observed under a stereomicroscope (SDPTOP SZN71). Five replicates were taken for each treatment, and 5 explants were planted in each culture dish or flask. Data represented the mean ± standard error of the replicates. One-way analysis of variance (ANOVA) for data evaluation and Duncan’s multiple range test [42] with a 0.05 level of probability were performed using the SPSS PASW Statistics 18.0 software (IBM, Chicago, IL, USA).

## Figures and Tables

**Figure 1 plants-10-00067-f001:**
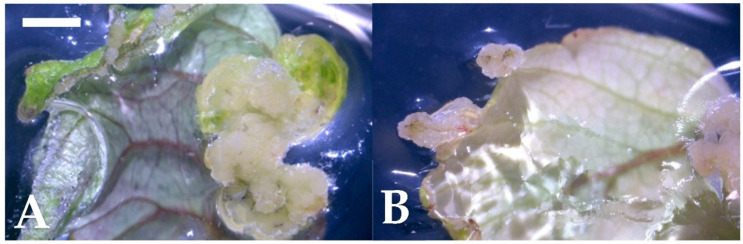
Callus induction from the leaf culture of strawberry after a 4-week culture (bar). (**A**) Thidiazuron (TDZ) 2.27 µM and 2,4-dichlorophenoxyacetic acid (2,4-D) 0.45 µM combination (1 mm); (**B**) TDZ 4.54 µM and 2,4-D 2.27 µM combination (1 mm).

**Figure 2 plants-10-00067-f002:**
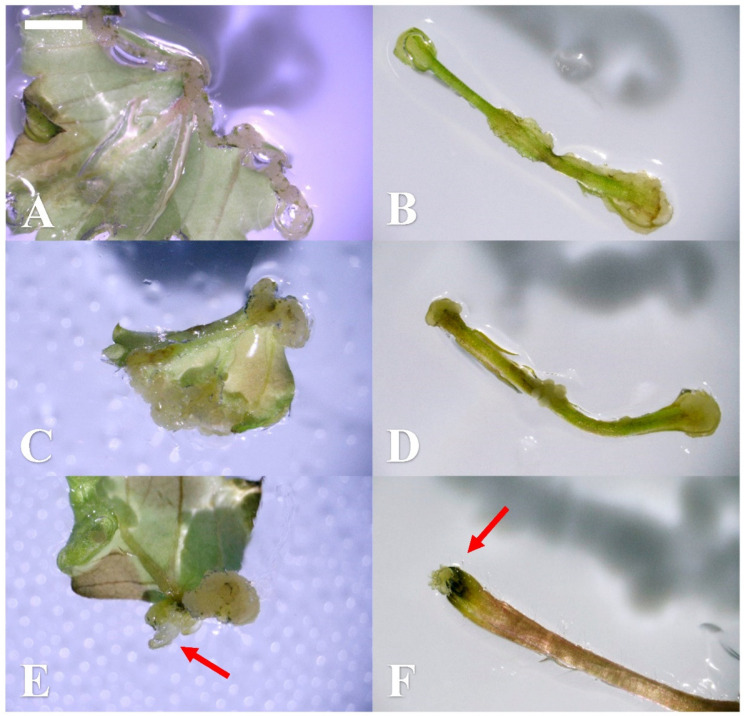
Callus induction and shoot regeneration using different explants (bar). After a 2-week culture, calli were found on (**A**) the whole leaf and (**B**) the petiole explants under the thidiazuron (TDZ) 4.53 µM and 2,4-dichlorophenoxyacetic acid (2,4-D) 2.27 µM combination (1 mm) in the dark condition. After a 4-week culture, both calli and shoots were observed. More calli were seen in (**C**) the whole leaf and (**D**) the petiole explants culture under the TDZ 2.27 µM and 2,4-D 2.27 µM combination (1 mm). The shoots started to develop in (**E**) the whole leaf and (**F**) the petiole explants culture under the TDZ 4.54 µM and 2,4-D 0.45 µM combination (1 mm).

**Figure 3 plants-10-00067-f003:**
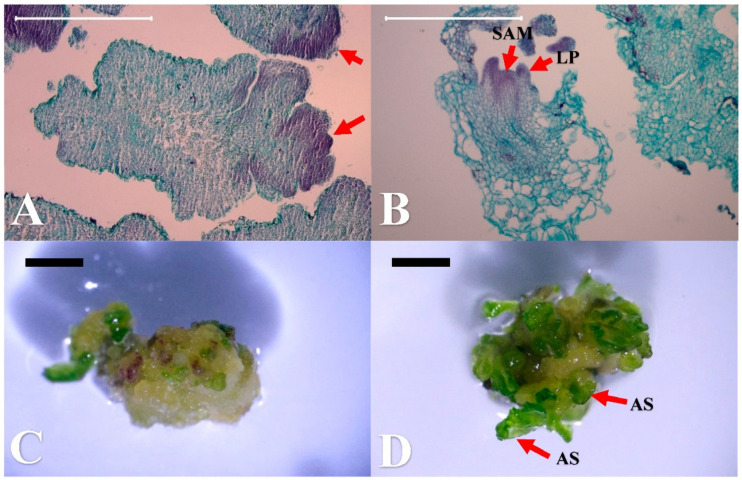
Histological observation of the shoot regeneration from wild strawberry (*Fragaria vesca*) calli after 2 weeks of culturing, with their corresponding development stages (bar). (**A**) Meristematic cells originated from the cells of the callus explants (500 μm); (**B**) shoot organogenesis with leaf primordium (LP) and shoot apical meristem (SAM) (500 μm); (**C**) the corresponding developmental stage of the callus explants corresponding to (A) (1 mm); (**D**) the stage of adventitious shoot (AS) development corresponding to (B) (1 mm).

**Figure 4 plants-10-00067-f004:**
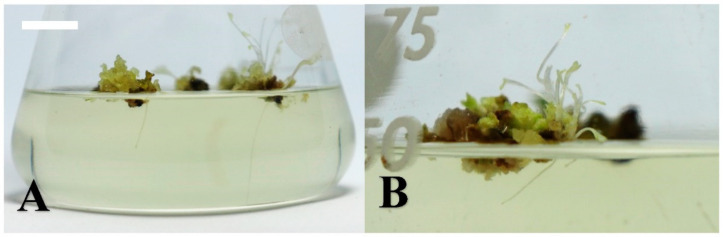
Root induction from wild strawberry (*Fragaria vesca*) shoots under the dark condition after 8 weeks of culturing (bar). (**A**) the indole-3-butyric acid (IBA) 0.41 µM treatment (10 mm); (**B**) the IBA 4.14 µM treatment (5 mm).

**Figure 5 plants-10-00067-f005:**
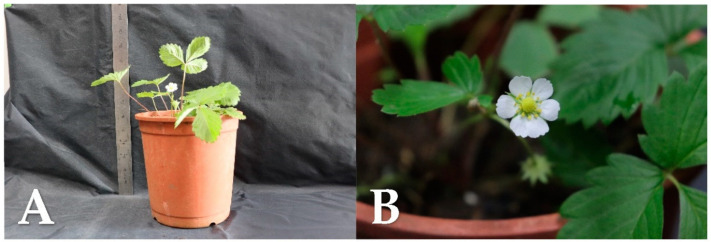
Acclimatization of wild strawberry (*Fragaria vesca*) plants. (**A**) The strawberry after three months of cultivation; (**B**) flowers blooming after strawberry cultivation.

**Table 1 plants-10-00067-t001:** Effects of the combination of thidiazuron (TDZ) or 6-benzylaminopurine (BA) plus 2,4-dichlorophenoxyacetic acid (2,4-D) on strawberry (*Fragaria vesca*) leaf explants after 4 weeks of culturing.

PGR Combination (µM)			Callus Induction Rate (%)				Browning Rate (%)			
TDZ	BA	2,4-D	Dark		Light		Dark		Light	
0		0	0 ± 0	e ^1^	0 ± 0	e	12 ± 8	bc	100 ± 0	a
0		0.45	0 ± 0	e	0 ± 0	e	96 ± 4	a	100 ± 0	a
0		2.27	0 ± 0	e	0 ± 0	e	4 ± 4	c	0 ± 0	d
0		4.53	0 ± 0	e	0 ± 0	e	100 ± 0	a	100 ± 0	a
0.45		0	48 ± 16	d	24 ± 10	e	4 ± 4	c	24 ± 10	bc
0.45		0.45	88 ± 8	abc	100 ± 0	a	0 ± 0	c	0 ± 0	d
0.45		2.27	92 ± 5	ab	92 ± 5	abc	0 ± 0	c	0 ± 0	d
0.45		4.53	92 ± 5	ab	96 ± 5	ab	0 ± 0	c	0 ± 0	d
2.27		0	68 ± 16	bcd	16 ± 12	e	0 ± 0	c	4 ± 4	d
2.27		0.45	96 ± 4	a	100 ± 0	a	0 ± 0	c	0 ± 0	d
2.27		2.27	100 ± 0	a	100 ± 0	a	0 ± 0	c	0 ± 0	d
2.27		4.53	96 ± 4	a	100 ± 0	a	0 ± 0	c	0 ± 0	d
4.54		0	0 ± 0	e	16 ± 7	e	28 ± 19	b	0 ± 0	d
4.54		0.45	76 ± 19	abc	100 ± 0	a	4 ± 4	c	0 ± 0	d
4.54		2.27	96 ± 4	a	84 ± 16	abc	4 ± 4	c	0 ± 0	d
4.54		4.53	100 ± 0	a	80 ± 20	abc	0 ± 0	c	20 ± 20	bcd
	2.22	0	0 ± 0	e	0 ± 0	e	4 ± 4	c	100 ± 0	a
	2.22	0.45	0 ± 0	e	0 ± 0	e	28 ± 19	b	100 ± 0	a
	2.22	2.27	76 ± 10	abc	0 ± 0	e	0 ± 0	c	100 ± 0	a
	2.22	4.53	100 ± 0	a	92 ± 5	abc	0 ± 0	c	4 ± 4	d
	4.44	0	20 ± 11	e	0 ± 0	e	0 ± 0	c	80 ± 13	a
	4.44	0.45	16 ± 12	e	8 ± 5	e	12 ± 8	bc	24 ± 7	bc
	4.44	2.27	84 ± 4	abc	72 ± 10	c	0 ± 0	c	0 ± 0	d
	4.44	4.53	64 ± 12	cd	52 ± 15	d	0 ± 0	c	36 ± 17	b
	8.88	0	0 ± 0	e	0 ± 0	e	16 ± 10	bc	100 ± 0	a
	8.88	0.45	4 ± 4	e	0 ± 0	e	8 ± 5	bc	100 ± 0	a
	8.88	2.27	80 ± 9	abc	76 ± 7	bc	0 ± 0	c	16 ± 7	cd
	8.88	4.53	68 ± 10	bcd	0 ± 0	e	0 ± 0	c	100 ± 0	a

Note: (1) Means ± standard error of 5 replicates with the same letters are not significantly different at *p* < 0.05; (2) the cultures were in the dark or under a 16:8 h photoperiod at 3500 lux.

**Table 2 plants-10-00067-t002:** Effect of the thidiazuron (TDZ) and 2,4-dichlorophenoxyacetic acid (2,4-D) combination on the callus induction rate from different strawberry (*Fragaria vesca*) explants after 2 weeks of culturing.

PGR Combination (µM)		Whole Leaf (%)		Tip Blade (%)		Basal Blade (%)		Petiole (%)	
TDZ	2,4-D	Dark	Light	Dark	Light	Dark	Light	Dark	Light
0	0	0 ± 0 h ^1^	0 ± 0 c	0 ± 0 e	0 ± 0 b	0 ± 0 c	0 ± 0 d	0 ± 0 c	0 ± 0 b
0	0.45	0 ± 0 h	0 ± 0 c	0 ± 0 e	0 ± 0 b	0 ± 0 c	0 ± 0 d	0 ± 0 c	0 ± 0 b
0	2.27	4 ± 4 gh	12 ± 8 c	0 ± 0 e	0 ± 0 b	0 ± 0 c	0 ± 0 d	4 ± 4 c	0 ± 0 b
0.45	0	24 ± 7 fg	5 ± 4 c	8 ± 5 e	0 ± 0 b	28 ± 10 b	0 ± 0 d	4 ± 4 c	0 ± 0 b
0.45	0.45	52 ± 10 de	16 ± 12 c	72 ± 10 bc	32 ± 16 b	84 ± 12 a	28 ± 20 cd	4 ± 4 c	0 ± 0 b
0.45	2.27	40 ± 13 ef	64 ± 12 ab	80 ± 6 ab	70 ± 10 a	88 ± 5 a	72 ± 8 ab	12 ± 8 c	8 ± 5 ab
2.27	0	32 ± 10 ef	24 ± 12 c	12 ± 5 e	0 ± 0 b	36 ± 12 b	4 ± 4 d	8 ± 5 c	0 ± 0 b
2.27	0.45	76 ± 7 bc	80 ± 9 ab	96 ± 4 a	32 ± 12 b	80 ± 13 a	30 ± 10 cd	36 ± 10 b	4 ± 4 b
2.27	2.27	100 ± 0 a	80 ± 9 ab	16 ± 7 e	76 ± 15 a	72 ± 10 a	84 ± 12 a	88 ± 8 a	16 ± 7 a
4.54	0	0 ± 0 h	25 ± 10 c	56 ± 13cd	8 ± 5 b	28 ± 5 b	12 ± 8 d	5 ± 4 c	4 ± 4 b
4.54	0.45	92 ± 8 ab	88 ± 5 a	40 ± 13 d	32 ± 20 b	84 ± 10 a	48 ± 21 bc	44 ± 7 b	12 ± 5 a
4.54	2.27	64 ± 10 cd	60 ± 9 b	40 ± 6 d	30 ± 13 b	76 ± 7 a	80 ± 11 a	48 ± 12 b	16 ± 12 a

Note: (1) Means ± standard error of 5 replicates with the same letters are not significantly different at *p* < 0.05; (2) the cultures were in the dark or under a 16:8 h photoperiod at 3500 lux.

**Table 3 plants-10-00067-t003:** Effect of the thidiazuron (TDZ) and 2,4-dichlorophenoxyacetic acid (2,4-D) combination on the callus induction rate from different strawberry (*Fragaria vesca*) explants after 4 weeks of culturing.

PGR Combination (µM)		Whole Leaf (%)		Tip Blade (%)		Basal Blade (%)		Petiole (%)	
TDZ	2,4-D	Dark	Light	Dark	Light	Dark	Light	Dark	Light
0	0	0 ± 0 c ^1^	0 ± 0 c	0 ± 0 f	0 ± 0 b	4 ± 4 c	0 ± 0 e	0 ± 0 c	0 ± 0 d
0	0.45	0 ± 0 c	0 ± 0 c	20 ± 6 ef	0 ± 0 b	0 ± 0 c	0 ± 0 e	0 ± 0 c	0 ± 0 d
0	2.27	12 ± 8 c	20 ± 13 c	4 ± 4 f	4 ± 4 b	4 ± 4 c	0 ± 0 e	40 ± 9 b	4 ± 4 d
0.45	0	44 ± 13 c	20 ± 6 c	8 ± 5 ef	0 ± 0 b	40 ± 9 b	0 ± 0 e	0 ± 0 c	0 ± 0 d
0.45	0.45	92 ± 5 a	24 ± 19 c	80 ± 6 ab	60 ± 11 a	92 ± 5 a	40 ± 18 d	32 ± 14 b	4 ± 4 d
0.45	2.27	84 ± 7 a	92 ± 5 a	95 ± 4 a	70 ± 10 a	88 ± 8 a	84 ± 10 ab	80 ± 6 a	72 ± 14 bc
2.27	0	48 ± 10 c	24 ± 7 c	32 ± 14 de	4 ± 4 b	40 ± 14 b	8 ± 5 e	4 ± 4 c	8 ± 8 d
2.27	0.45	88 ± 8 a	88 ± 8 a	96 ± 4 a	68 ± 16 a	100 ± 0 a	55 ± 10 cd	92 ± 8 a	24 ± 12 d
2.27	2.27	100 ± 0 a	96 ± 4 a	48 ± 12 cd	80 ± 13 a	96 ± 4 a	100 ± 0 a	100 ± 0 a	100 ± 0 a
4.54	0	4 ± 4 c	55 ± 13 b	60 ± 14 bc	16 ± 7 b	40 ± 11 b	12 ± 8 e	10 ± 4 c	8 ± 8 d
4.54	0.45	100 ± 0 a	96 ± 4 a	48 ± 10 cd	48 ± 22 a	100 ± 0 a	68 ± 16 bc	92 ± 5 a	60 ± 14 c
4.54	2.27	84 ± 7 a	72 ± 7 ab	60 ± 6 bc	55 ± 10 a	84 ± 7 a	95 ± 4 a	84 ± 7 a	84 ± 7 ab

Note: (1) Means ± standard error of 5 replicates with the same letters are not significantly different at *p* < 0.05; (2) the cultures were in the dark or under a 16:8 h photoperiod at 3500 lux.

**Table 4 plants-10-00067-t004:** Effect of the thidiazuron (TDZ) and 2,4-dichlorophenoxyacetic acid (2,4-D) combination on the shoot regeneration rate from different strawberry (*Fragaria vesca*) explants after 4 weeks of culturing.

PGR Combination (µM)		Whole Leaf (%)		Tip Blade (%)		Basal Blade (%)		Petiole (%)	
TDZ	2,4-D	Dark	Light	Dark	Light	Dark	Light	Dark	Light
0	0	0 ± 0	0 ± 0 b ^1^	0 ± 0 b	0	0 ± 0 b	0	0 ± 0 b	0
0	0.45	0 ± 0	0 ± 0 b	0 ± 0 b	0	0 ± 0 b	0	0 ± 0 b	0
0	2.27	0 ± 0	0 ± 0 b	0 ± 0 b	0	0 ± 0 b	0	0 ± 0 b	0
0.45	0	0 ± 0	0 ± 0 b	0 ± 0 b	0	0 ± 0 b	0	0 ± 0 b	0
0.45	0.45	0 ± 0	0 ± 0 b	0 ± 0 b	0	0 ± 0 b	0	0 ± 0 b	0
0.45	2.27	0 ± 0	0 ± 0 b	0 ± 0 b	0	0 ± 0 b	0	0 ± 0 b	0
2.27	0	0 ± 0	0 ± 0 b	0 ± 0 b	0	4 ± 4 b	0	0 ± 0 b	0
2.27	0.45	12 ± 12	4 ± 4 a	0 ± 0 b	0	0 ± 0 b	0	4 ± 4 ab	0
2.27	2.27	4 ± 4	0 ± 0 b	0 ± 0 b	0	4 ± 4 b	0	0 ± 0 b	0
4.54	0	0 ± 0	0 ± 0 b	0 ± 0 b	0	0 ± 0 b	0	0 ± 0 b	0
4.54	0.45	12 ± 5	0 ± 0 b	8 ± 8 a	0	16 ± 7 a	0	8 ± 5 a	0
4.54	2.27	0 ± 0	0 ± 0 b	0 ± 0 b	0	0 ± 0 b	0	0 ± 0 b	0

Note: (1) Means ± standard error of 5 replicates with the same letters are not significantly different at *p* < 0.05; (2) the cultures were in the dark or under a 16:8 h photoperiod at 3500 lux.

**Table 5 plants-10-00067-t005:** Effect of the thidiazuron (TDZ) and 2,4-dichlorophenoxyacetic acid (2,4-D) combination on the number of shoot inductions from different strawberry (*Fragaria vesca*) explants after 4 weeks of culturing.

PGR Combination (µM)		Whole Leaf (Shoot No./Explant)		Tip Blade (Shoot No./Explant)		Basal Blade (Shoot No./Explant)		Petiole (Shoot No./Explant)	
TDZ	2,4-D	Dark	Light	Dark	Light	Dark	Light	Dark	Light
0	0	0 ± 0	0 ± 0 b ^1^	0 ± 0 b	0	0 ± 0 b	0	0 ± 0 b	0
0	0.45	0 ± 0	0 ± 0 b	0 ± 0 b	0	0 ± 0 b	0	0 ± 0 b	0
0	2.27	0 ± 0	0 ± 0 b	0 ± 0 b	0	0 ± 0 b	0	0 ± 0 b	0
0.45	0	0 ± 0	0 ± 0 b	0 ± 0 b	0	0 ± 0 b	0	0 ± 0 b	0
0.45	0.45	0 ± 0	0 ± 0 b	0 ± 0 b	0	0 ± 0 b	0	0 ± 0 b	0
0.45	2.27	0 ± 0	0 ± 0 b	0 ± 0 b	0	0 ± 0 b	0	0 ± 0 b	0
2.27	0	0 ± 0	0 ± 0 b	0 ± 0 b	0	0.2 ± 0.2 b	0	0 ± 0 b	0
2.27	0.45	0.8 ± 0.80	0.2 ± 0.2 a	0 ± 0 b	0	0 ± 0 b	0	0.2 ± 0.2 b	0
2.27	2.27	0.4 ± 0.40	0 ± 0 b	0 ± 0 b	0	0.2 ± 0.2 b	0	0 ± 0 b	0
4.54	0	0 ± 0	0 ± 0 b	0 ± 0 b	0	0 ± 0 b	0	0 ± 0 b	0
4.54	0.45	0.6 ± 0.24	0 ± 0 b	0.4 ± 0.4 a	0	1.0 ± 0.45 a	0	0.6 ± 0.4 a	0
4.54	2.27	0 ± 0	0 ± 0 b	0 ± 0 b	0	0 ± 0 b	0	0 ± 0 b	0

Note: (1) Means ± standard error of 5 replicates with the same letters are not significantly different at *p* < 0.05; (2) the cultures were in the dark or under a 16:8 h photoperiod at 3500 lux.

**Table 6 plants-10-00067-t006:** Effect of thidiazuron (TDZ), 6-benzylaminopurine (BA) and kinetin (KIN) on shoot regeneration from strawberry (*Fragaria vesca*) callus explants after 2 weeks of culturing.

Cytokinin (µM)			No. of Inducted Shoots		Shoot Induction Rate (%)		Browning Rate (%)	
TDZ	BA	KIN	Dark	Light	Dark	Light	Dark	Light
0	0	0	1.2 ± 0.6 cd ^1^	9.2 ± 1.5 bc	16 ± 7 de	64 ± 4 abcd	0	0
0.45			9.4 ± 1.3 a	11.6 ± 1.6 b	92 ± 5 a	88 ± 5 ab	0	0
2.27			4.4 ± 0.9 b	9.0 ± 3.5 bc	56 ± 12 b	56 ± 16 bcd	0	0
4.54			1.0 ± 0.6 d	3.8 ± 2.1 c	12 ± 5 de	36 ± 19 d	0	0
9.08			2.4 ± 1.0 bcd	4.2 ± 0.8 c	36 ± 12 bcd	64 ± 12 abcd	0	0
	0.44		8.0 ± 1.3 a	19.4 ± 1.4 a	88 ± 8 a	96 ± 4 a	0	0
	2.22		0.4 ± 0.2 d	4.6 ± 0.5 c	8 ± 5 e	48 ± 5 cd	0	0
	4.44		0.4 ± 0.2 d	4.8 ± 1.0 c	8 ± 5 e	48 ± 10 cd	0	0
	8.88		1.6 ± 0.7 cd	4.4 ± 1.7 c	20 ± 8 cde	56 ± 20 bcd	0	0
		0.47	4.4 ± 0.9 b	9.0 ± 1.3 bc	48 ± 10 bc	80 ± 6 abc	0	0
		2.33	2.0 ± 1.0 bcd	13.8 ± 3.5 ab	24 ± 10 cde	88 ± 8 ab	0	0
		4.65	2.2 ± 0.9 bcd	12.2 ± 2.6 b	24 ± 7 cde	88 ± 12 ab	0	0
		9.30	3.8 ± 0.4 bc	9.4 ± 1.6 bc	60 ± 6 b	80 ± 9 abc	0	0

Note: (1) Means ± standard error of 5 replicates with the same letters are not significantly different at *p* < 0.05; (2) the cultures were under a 16:8 h photoperiod at 3500 lux.

**Table 7 plants-10-00067-t007:** Effect of indole-3-butyric acid (IBA) and 1-naphthaleneacetic acid (NAA) on the root induction from strawberry (*Fragaria vesca*) shoot explants after 8 weeks of culturing.

Auxin (µM)		No. of Induced Roots		Root induction Rate (%)		Browning Rate (%)	
IBA	NAA	Dark	Light	Dark	Light	Dark	Light
0		0.0 ± 0	0.0 ± 0	0 ± 0 b ^1^	0 ± 0	0 ± 0	20 ± 6
0.41		0.6 ± 0.4	0.0 ± 0	12 ± 8 a	0 ± 0	4 ± 4	12 ± 8
1.03		0.0 ± 0	0.0 ± 0	0 ± 0 b	0 ± 0	0 ± 0	20 ± 6
2.07		0.0 ± 0	0.0 ± 0	0 ± 0 b	0 ± 0	0 ± 0	8 ± 4
4.14		0.6 ± 0.4	0.0 ± 0	8 ± 5 ab	0 ± 0	0 ± 0	12 ± 8
8.28		0.4 ± 0.4	0.0 ± 0	4 ± 4 ab	0 ± 0	0 ± 0	12 ± 12
	0.48	0.0 ± 0	0.2 ± 0.2	0 ± 0 b	4 ± 4	12 ± 12	8 ± 5
	1.20	0.0 ± 0	0.2 ± 0.2	0 ± 0 b	4 ± 4	4 ± 4	0 ± 0
	2.40	0.0 ± 0	0.0 ± 0	0 ± 0 b	0 ± 0	0 ± 0	16 ± 7
	4.80	0.0 ± 0	0.0 ± 0	0 ± 0 b	0 ± 0	4 ± 4	20 ± 9
	9.60	0.0 ± 0	0.0 ± 0	0 ± 0 b	0 ± 0	8 ± 8	12 ± 5

Note: (1) Means ± standard error of 5 replicates with the same letters are not significantly different at *p* < 0.05; (2) the cultures were under a 16:8 h photoperiod at 3500 lux.

## Data Availability

The data presented in this study are available on request from the corresponding author.

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
