# Peer review of "Use of Thidiazuron for High-Frequency Callus Induction and Organogenesis of Wild Strawberry (Fragaria vesca)"

_plants, 2020, doi:10.3390/plants10010067_

Round 1
Reviewer 1 Report
This paper is evaluated to contain some novel findings related to callus induction and plant regeneration from callus in wild strawberry. However, there are many parts to be modified in the manuscript. Please correct the manuscript according to the comments in the attached file. Especially, concentration of PGR is required to be expressed as mol/L.

Author Response
Response: We thank the reviewer for these important comments. We really appreciate that reviewer pointed out so many mistakes in the article. All these comments have been revised according to the reviewer’s suggestions which is a great help for us. Regarding the concentration of PGR, we have modified the units into µM. Just wondering if you feel okay with the revision. Please check the attached file of the revised manuscript version.

Reviewer 2 Report
This research lacks innovation. There are scientific articles regarding the in vitro culture of Fragaria vesca that the authors fail to mention. Below there are a few of them:
Yildirim, A.B.; Turker, A.U. Effects of regeneration enhancers on micropropagation of Fragaria vesca L. and phenolic content comparison of field-grown and in vitro-grown plant materials by liquid chromatography-electrospray tandem mass spectrometry LC–ESI-MS/MS. Sci. Hortic. 2014, 169, 169–178.
Landi, L.; Mezzetti, B. TDZ, auxin and genotype effects on leaf organogenesis in Fragaria. Plant Cell Rep. 2006, 25, 281–288
Babul C. Sarker, Douglas D. Archbold, Robert L. Geneve and Sharon T. Kester. Rapid In Vitro Multiplication of Non-Runnering Fragaria vesca Genotypes from Seedling Shoot Axillary Bud Explants. Horticulturae 2020, 6, 51; doi:10.3390/horticulturae6030051
Author Response
Response: We really thank the comments from the reviewer and also provide some more scientific publications. Indeed, our work has some similar experimental designs as the publications you provided. Compared to Yildirim and Turker used leaves and petioles as explants, we also test the different parts of the leaf. Regarding the effect of PGRs, we found that TDZ could induce the calli with 2,4-D in very low concentration (0.1 mg/L). We also tested the effect of light or dark. Landi and Mezzetti did test the difference under the light period and dark. However, they more focused on different genotypes of strawberries rather than explants and PGRs. Sarker et al. applied shoot axillary bud and leaf explants. The 4.6 shoots could be successfully induced from leaf explants after 9 weeks. Most of these reports showed that the explants put on the same medium for observation of their morphogenesis. We did observe and know that callus induction and shoot formation can happen together on the same medium. However, in our present work, we want more carefully to divide the morphogenesis into callus induction, shoot organogenesis, and root formation stages and to characterize the suitable induced factors for each stage. In our study, 100% of calli can be induced from leaves after 2 weeks in the dark, and then, calli were transferred to BA 0.1 mg/L medium. After another 2-weeks culture in the light condition, 19.4 shoots can be obtained from each explant. We optimized the PGRs and environmental factors and shortened the regeneration period. I think it is one of the important in our work. We really appreciate your valuable comments. The papers that you provided will be also put into our reference. Please check the attached file of the revised manuscript version.

Reviewer 3 Report
The manuscript is well structured and well written and deserves to be published with some minor corrections indicated in the attached file. I personally checked the reliability of the data and the feedback was positive. Some references to add in the introduction, the conclusions to be completed, the collection sites of the individuals used in the laboratory is not clear. It is necessary to move the materials and method before the results.

Author Response
Response: We really appreciate your kind comments. We do really look forward to publishing this study. We worked carefully according to your suggestion. I have added some references to the induction part. In conclusion, I add a few sentences and reference to make it more complete. The description of collected explants has been improved in the materials and methods. Regarding the move of materials and methods part, the structure of the manuscript is based on the journal's template. Therefore, the assistant editor still recommended putting the material method behind. Please check the attached file of the revised manuscript version. Thank you so much again for your valuable comments and help.

Reviewer 4 Report
Title
I recommend you review your title. I just suggest " Use of TDZ for high-frequency callus induction and organogenesis of wild strawberry (Fragaria vesca)"
Introduction
I suggest reviewing, the organization and structure of the introduction. I just suggest that the introduction ought to address the importance of the use of the in vitro technique for the production of true-to-type certified vegetative material and for the application of advanced biotechnologies for the genetic improvement of berries, including the genetic transformation techniques.
Line 39-181 less information, it is not a paper about nutritional composition of strawberry.
Line 197 Please remove "Today"
Line 197-203 Is redundant, please add this part at the beginning of Introduction (Line 33-38)
Results
I suggest reviewing, the organization and structure of the results. The results not explained clearly the experiment.
-Effect of genotype on leaf response-Effects of PGRs on leaf differentiation
-Effects of light conditions on leaf differentiation
Add table with effects of PGR combinations on the number of shoots from the regenerating leaves of cultivars (Leaves with calli; Leaves with shoots; n◦ shoots/leaves).
Materials and Methods
Line 1109 Please remove "the biosafety cabinet"
Line 1425 Please change with " a complete randomization of the PGR combinations"
Author Response
Point-by-point responses to the reviewer's comments for Manuscript ID: plants-1002865
Title
I recommend you review your title. I just suggest "Use of TDZ for high-frequency callus induction and organogenesis of wild strawberry (Fragaria vesca)"
Response: We really appreciate your kind suggestions. I have revised it to "Use of Thidiazuron for High-Frequency Callus Induction and Organogenesis of Wild Strawberry (Fragaria vesca)" as your suggestion. Thank you! Please check it in the attached file.
Introduction
I suggest reviewing, the organization and structure of the introduction. I just suggest that the introduction ought to address the importance of the use of the in vitro technique for the production of true-to-type certified vegetative material and for the application of advanced biotechnologies for the genetic improvement of berries, including the genetic transformation techniques.
Response: We really appreciate your kind suggestion! We have revised the introduction part as your suggestion and emphasized the concept that you suggested. Thank you very much! Please check it in the attached file.
Line 39-181 less information, it is not a paper about the nutritional composition of strawberries.
Response: We appreciate your suggestion. I have reduced the information and combined it with the text in line 197 to 203 as our new second paragraph. Please check it in the attached file.
Line 197 Please remove "Today"
Response: Thank you! I have deleted it.
Line 197-203 Is redundant, please add this part at the beginning of Introduction (Line 33-38)
Response: Thank you so much for the suggestion! I have moved this part to the second paragraph. Please check it in the attached file.
Results
I suggest reviewing, the organization and structure of the results. The results not explained clearly the experiment.
Response: Dear reviewer, Thanks so much for your comments! Please check the revised version in the attached file. I have added more description in the text and try to improve and explain clearly in my result part. Please check it in the attached file.
-Effect of genotype on leaf response
Response: Dear reviewer, thanks for your comments. Indeed, the different genotypes can be affected by the response. However, I did not characterize the different genotypes in this study. I will include this important factor in future studies. Thanks again!
-Effects of PGRs on leaf differentiation
Response: Dear reviewer, thanks for your comments! Regarding leaf differentiation, the culture of leaf explants can be found in sections 2.1 and 2.2. I have revised and improved the description in these two sections. Please check it in the attached file.
-Effects of light conditions on leaf differentiation
Response: Dear reviewer, thanks for your comments! Basically, the light condition did not benefit for callus induction but accelerated the shoot regeneration which has been written in our result and discussion. Please check it in the attached file.
Add table with effects of PGR combinations on the number of shoots from the regenerating leaves of cultivars (Leaves with calli; Leaves with shoots; n◦ shoots/leaves).
Response: Dear reviewer, thanks for your comments! I have added a new table in the text (new table 5) which exhibited the number of shoot formation per explant (leaf). Regarding the leaves with calli or shoots, we utilized the percentage to present the data in Table 1-4. Each treatment contained 5 replications and 1 replication had 5 explants. If 1 leaf explant with callus within 5 explants, the replication will show as 20% induction rate. Therefore, our data actually is the same as that you suggested (leaves with calli or shoots) but presented in a different way. I really appreciate your kind suggestions!
Materials and Methods
Line 1109 Please remove "the biosafety cabinet"
Response: Thank you! I have deleted it.
Line 1425 Please change with " a complete randomization of the PGR combinations"
Response: Thank you so much! I have revised it.

Round 2
Reviewer 2 Report
The revised version of the manuscript has been improved. However, it still lacks originality.
Reviewer 4 Report
Line 586 Please change with "combination"